# Modulating Expression of Thioredoxin Interacting Protein (TXNIP) Prevents Secondary Damage and Preserves Visual Function in a Mouse Model of Ischemia/Reperfusion

**DOI:** 10.3390/ijms20163969

**Published:** 2019-08-15

**Authors:** Maha Coucha, Ahmed Y. Shanab, Mohamed Sayed, Almira Vazdarjanova, Azza B. El-Remessy

**Affiliations:** 1Augusta Biomedical Research Corporation, Augusta, GA 30901, USA; 2Charlie Norwood VA Medical Center, Augusta, GA 30904, USA; 3Department of Pharmaceutical Sciences, South University, School of Pharmacy, Savannah, GA 31406, USA; 4Department of Pharmacology and Toxicology, Augusta University, Augusta, GA 30901, USA; 5Department of Pharmacy, Doctors Hospital of Augusta, Augusta, GA 30909, USA

**Keywords:** TXNIP, retinal inflammation, inflammasome, ischemia reperfusion, visual function

## Abstract

Retinal neurodegeneration, an early characteristic of several blinding diseases, triggers glial activation, resulting in inflammation, secondary damage and visual impairment. Treatments that aim only at neuroprotection have failed clinically. Here, we examine the impact of modulating thioredoxin interacting protein (TXNIP) to the inflammatory secondary damage and visual impairment in a model of ischemia/reperfusion (IR). Wild type (WT) and TXNIP knockout (TKO) mice underwent IR injury by increasing intraocular pressure for 40 min, followed by reperfusion. An additional group of WT mice received intravitreal TXNIP-antisense oligomers (ASO, 100 µg/2 µL) 2 days post IR injury. Activation of Müller glial cells, apoptosis and expression of inflammasome markers and visual function were assessed. IR injury triggered early TXNIP mRNA expression that persisted for 14 days and was localized within activated Müller cells in WT-IR, compared to sham controls. Exposure of Müller cells to hypoxia-reoxygenation injury triggered endoplasmic reticulum (ER) stress markers and inflammasome activation in WT cells, but not from TKO cells. Secondary damage was evident by the significant increase in the number of occluded acellular capillaries and visual impairment in IR-WT mice but not in IR-TKO. Intervention with TXNIP-ASO prevented ischemia-induced glial activation and neuro-vascular degeneration, and improved visual function compared to untreated WT. Targeting TXNIP expression may offer an effective approach in the prevention of secondary damage associated with retinal neurodegenerative diseases.

## 1. Introduction

Retinal neurodegeneration is an early and common characteristic of several blinding diseases, including diabetic retinopathy, retinopathy of prematurity, traumatic optic neuropathy and glaucoma [1]. Despite the significant progress to develop therapeutics for the treatment of the proliferative stage of ischemic retinopathy, the underlying cause of ischemia/hypoxia remain unresolved, resulting in progression of the disease [2]. On the other hand, treatment strategies that targeted only neuroprotection in optic neuropathy failed clinically (reviewed in [3]). Neurodegeneration triggers glial activation, resulting in inflammation and secondary damage to other retina cell types, which sustain cellular damage and eventually cause visual impairment [4]. Thus, there is a great need to identify new therapeutics that can prevent secondary damage of the retina in a practical window and effectively prevent vision loss associated with these ischemic retinal diseases.

Exposure to ischemia or traumatic compression can compromise retinal antioxidant defenses, resulting in a significant increases in retinal oxidative stress [5,6]. Thioredoxin, a major antioxidant system, is negatively regulated by the thioredoxin interacting protein (TXNIP). Expression of TXNIP is known to be directly regulated by glucose level and Ca^2+^ influx and is immediately up-regulated in neurons in response to ischemic insult [7,8]. Additional mechanisms have been proposed to fine-tune expression of TXNIP, including microRNA, small non-coding RNAs that control the translation and transcription of various genes. We and others have shown that miR-17-5p is a regulator of TXNIP expression in pancreatic cells [9], Müller cells [10] and neural stem cells [11]. Destabilization of miR-17-5p has been closely linked to endoplasmic reticulum (ER) stress. Exposure to hypoxia, acidosis or depletion of calcium stores in the lumen of ER can disturb homeostasis and thus negatively impact protein-folding processes, leading to an accumulation of unfolded or misfolded proteins (reviewed in [12]). Unfolded protein response (UPR), an adaptive response that prevents the accumulation of misfolded proteins, is transduced by three major ER-resident stress sensors, namely protein kinase RNA-like ER kinase (PERK), activating transcription factor 6 (ATF6) and inositol requiring enzyme 1 (IRE-1) [13]. Among UPR pathways, IRE1α, an ER bifunctional kinase/RNase has been shown to destabilize the number of RNA and microRNA, including miR-17-5p in pancreatic beta cells [9] and Müller cells [10]. However, whether ocular ischemia can induce ER-stress and dysregulate miR-17-5p to sustain TXNIP expression remains unknown.

Recently, TXNIP has been identified as a direct activator of the NOD-like receptor pyrin domain containing 3 (NLRP3) inflammasome in various types of cells (reviewed in [14]). Increases in oxidative stress dissociate TXNIP from thioredoxin, allowing it to bind to NLRP3, resulting in its activation sustaining a vicious cycle of oxidative stress and inflammation. Our recent studies in high-fat models demonstrated for the first time that TXNIP is required for retinal inflammasome activation and release of IL-β [15,16]. Nevertheless, whether there is a link between ER-stress and sustained TXNIP-mediated retinal inflammation and secondary damage remains poorly understood. Here, we attempted to decipher the underlying mechanisms, and we tested the hypothesis that ER-stress sustains TXNIP mRNA expression via dysregulating miR-17-5p, resulting in retinal inflammation in response to ocular ischemia. Further, we examined the therapeutic utility of TXNIP antisense oligomers (ASO) as intervention therapy to prevent TXNIP-mediated secondary damage and delay or prevent vision impairment.

## 2. Results

### 2.1. Ocular Ischemia-Triggered Oxidative Stress and TXNIP Expression That Was Sustained for 14 Days

Exposure to ocular ischemia reperfusion (IR) injury triggered a significant increase in oxidative stress, evident by increased retinal lipid peroxide 4-hydroxyneonal (4-HNE) expression when compared to sham controls after 1 day of IR injury (Figure 1a). This effect was associated with a marked increase in TXNIP protein expression (Figure 1b). In parallel to this, IR injury triggered TXNIP mRNA levels (7-fold) when compared to sham controls (Figure 1c). Interestingly, the increase in TXNIP mRNA expression remained significantly higher (2–3 fold) at 3 and 14 days post-IR injury when compared to sham controls (Figure 1c).

### 2.2. Ocular Ischemia-Triggered Activation of Glial Müller Cells That Was Sustained for 14 Days

IR injury-induced Müller cell activation was assessed by intense radial staining of glial fibrillary acidic protein (GFAP) when compared to sham controls after 3 days (Figure 2a). Müller cell activation remained, although to a lesser extent after 14 days, which signifies the importance of the glial activation in response to a transient ischemic event. Next, we examined the expression of TXNIP in retina sections. As shown in Figure 2b, IR injury triggered TXNIP expression compared to shams. TXNIP (green) was markedly co-localized (yellow) within Müller cells stained with glutamine synthetase (GS, red) in retinas from IR mice when compared to retinas from sham mice.

### 2.3. IR Injury and Hypoxia-Triggered ER Stress Markers and Suppressed miR-17-5p Expression

#### 2.3.1. Hypoxia-Triggered ER Stress Markers and Suppressed miR-17-5p Expression In-Vitro

To better understand the specific role of glia in ER stress-mediated miR-17-5p and TXNIP regulation, rat Müller cells (rMC-1) were used. To mimic IR-injury, rMC-1 cells were exposed to hypoxia for 1 h, followed by reperfusion in normoxia for 24 h. Exposure to transient hypoxia, followed by reoxygenation, significantly upregulated (~2-fold) the ER stress marker inositol-requiring enzyme-1-alpha (IRE-1α) mRNA expression when compared to normoxia (Figure 3a). This effect coincided with significant decreases in expression of miR-17-5p (Figure 3b) and significant increases (20-fold) in TXNIP mRNA (Figure 3c) in response to hypoxia, compared to cells grown in normoxia.

#### 2.3.2. Deletion of TXNIP Prevents IR Injury-Mediated ER Stress and Dysregulation of miR-17-5p In-Vivo

Next, we examined the impact of IR injury and TXNIP on expression of ER stress and miR-17-5p. Male C57Bl/J (WT) or TXNIP knockout (TKO) mice were subjected to retinal ischemia by increasing intraocular pressure for 50 min. Mice were sacrificed after 1 day. Transient ocular IR injury induced a significant upregulation of ER stress marker IRE-1α (Figure 3d), caused significant decreases in miR-17-5p expression (Figure 3e) and triggered TXNIP expression (Figure 3f) when compared to sham controls. Deletion of TXNIP prevented IR injury-mediated increase in IRE-1α expression and restored miR-5-p expression to a normal level. These results suggest a feedback loop of TXNIP expression to regulate ER stress.

### 2.4. IR Injury and Hypoxia-Triggered NLRP3 Inflammasome Activation and IL-1β Expression

#### 2.4.1. Hypoxia-Triggered NLRP3 Inflammasome Activation and IL-1β Expression In-Vitro

Exposure of primary Müller cells isolated from wild type (WT) mice to transient hypoxia (1 h) followed by reoxygenation significantly triggered NLRP3 inflammasome activation, evident by significant increases in expression of caspase-1, (1.5-fold) (Figure 4a,c) and IL-1β (1.8-fold) when compared to normoxia (Figure 4a,d). However, there was no significant change in NLRP3 protein expression (Figure 4a,b).

#### 2.4.2. Deletion of TXNIP-Prevented IR-Induced Expression of Inflammatory Mediators In-Vivo

In Vivo, transient ocular IR-injury induced an upregulation in NLRP3 expression in retinas from WT mice, but not in TKO mice, after 1 day of IR injury (Figure 4e,f). There was no difference in other components of NLRP3 inflammasome (data not shown). The activation of TXNIP-NLRP3 inflammasome was evident by increased retinal IL-1β expression in in WT mice but not TKO mice (Figure 4e,g). In parallel, retinal TNFα was significantly elevated in WT mice, but not TKO mice, after 1 day of IR injury (Figure 4e,h). Moreover, we found that primary Müller cells isolated from WT mice that were subjected to hypoxia/reoxygenation showed significant activation of TXNIP-NLRP3 inflammasome evident by release of cleaved IL-1β compared to normoxia. In contrast, primary cells isolated from TKO-mice that were subjected to hypoxia/reoxygenation did not show the activation of inflammasome, nor the release of IL-1β, compared to normoxia (Appendix A).

### 2.5. Deletion of TXNIP Prevented IR-Mediated Gliosis and Neuronal Cell Death Post-IR Injury

WT mice and TKO mice were subjected to ocular ischemia by increasing intraocular pressure for 50 min. Mice were sacrificed after 3 or 14 days of the insult. IR injury induced a significant increase in Müller cell activation in retinal ganglion cells (RGC), the inner nuclear layer (INL) and outer nuclear layer (ONL) in WT compared to shams. Deletion of TXNIP prevented IR-induced Müller activation after 3 days of ischemia (Figure 5a). Next, we examined cell death in retinal layers using terminal dUTP nick end-labeling (TUNEL) assay. As shown in Figure 5b, IR-injury showed numerous TUNEL+ve cells compared to shams in WT. Although TUNEL+ve cells were detected in the retina ganglion cell (RGC) layer and inner nuclear layer (INL), the majority of cells were observed in the outer nuclear layer (ONL), suggesting propagation of initial RGC damage. Deletion of TXNIP-mitigated IR-induced neurodegeneration after 3 days of ischemia is shown in Figure 5b. 

### 2.6. Deletion of TXNIP-Prevented IR-Mediated Neuro and Vascular Degeneration Post-IR Injury

After 3 and 14 days, retinas were collected and subjected to trypsin digest and stained to visualize retina microvasculature. After 3 days, a count of cells in the ganglion cell layer (GCL) showed significant reduction of neuronal cells in response to IR injury when compared to shams (Figure 6a,b). Deletion of TXNIP prevented the IR-mediated decrease in the ganglion cell count. After 3 days, there was no detectable number of occluded (acellular) capillaries (data not shown). IR injury caused a significant increase in the formation of occluded (acellular) capillaries, hallmark of retinal ischemia and secondary damage, when compared to sham controls (Figure 6c,d). In contrast, deletion of TXNIP prevented secondary damage evident by a significantly lower number of occluded capillaries.

### 2.7. Intervention Prevented Sustained Expression of TXNIP, ER-Stress and JNK Activation

WT mice were subjected to ocular ischemia, and after 48 h the mice were randomly assigned to receive TXNIP antisense oligomer (ASO) or scrambled oligomers via intravitreal injection. Mice were sacrificed after 14 days of the IR injury. First, we confirmed that ASO, a novel specific TXNIP inhibitor, blunted the increase of TXNIP mRNA after retinal IR (Figure 7a). Of note, ASO-WT tended to show a higher level of TXNIP mRNA when compared to Scr-WT; however, it did not reach statistical significance. Next, we examined the impact of TXNIP-ASO on the expression of the ER stress marker C/EBP homologous protein (CHOP). As shown in Figure 7b, western blot analysis showed that TXNIP-ASO blunted the increase in CHOP expression after retinal IR. As shown in Figure 7c, transient ocular ischemia triggered the phosphorylation of the apoptotic marker JNK when compared to shams. Interventional treatment with TXNIP-ASO significantly reduced JNK phosphorylation after retinal IR in WT.

### 2.8. Modulation of TXNIP Expression Improved Visual Function after IR Injury

Finally, we examined the therapeutic utility of TXNIP-ASO on preventing or delaying secondary damage and vision loss using an interventional regimen. WT mice went through IR surgery, then 48 h later, mice were randomized to receive TXNIP-ASO or scrambled oligomers. We found that intervention with TXNIP-ASO resulted in a significant decrease in the formation of occluded (acellular) capillaries (Figure 8a,b) when compared with controls that received scrambled oligomers. In an attempt to assess the impact of sustained TXNIP and its secondary damage on visual function, we adopted a visual cue test. This modified water maze involved training and testing mice on the ‘cue’ version of the Morris water maze task, as described previously [17]. Higher escape times indicate visual impairment. As shown in Figure 8c, we found that inhibiting TXNIP post-injury ameliorated the impact of IR injury on visual function by significantly reducing the time taken to reach the platform at 6 and 9 days after injury, but not at 3 days (Figure 8d).

## 3. Discussion

The main findings of this study are that ocular ischemia triggered and sustained both glial Müller activation and TXNIP expression for up to 14 days (Figure 1 and Figure 2). The increase in TXNIP expression was induced, at least in part, by ER stress and dysregulation of miR-17-5p in vivo and in Müller cell cultures (Figure 3). Interestingly, deletion of TXNIP prevented IR induced ER stress, and restored miR-17-5p to a normal level. Furthermore, TXNIP was required for hypoxia-induced inflammasome activation and the release of IL-1β, resulting in retinal inflammation (Figure 4). Deletion of TXNIP prevented glial activation, neuronal cell death, development of occluded capillaries in response to transient ischemia (Figure 5 and Figure 6). Intervention treatment with TXNIP-ASO significantly mitigated the long-term effects of IR-induced ER stress, microvascular degeneration and preserved visual function (Figure 7 and Figure 8). We believe that this is the first report that demonstrates that post-injury intervention of TXNIP expression can provide an effective therapeutic strategy to prevent secondary damage and vision loss in a model of IR (See Graphic Abstract).

Oxidative stress has been well-documented in ischemic retinopathy, including the IR model [18,19]. Our results showed that IR triggered significant increases in retinal TXNIP and oxidative stress, evident by increases in 4-HNE lipid peroxides. Our results lend further support to prior reports showing increases in retinal oxidative stress and lipid peroxidation [20,21]. Upregulation of TXNIP expression modulated the cellular redox state by inhibiting the thioredoxin system and shifting the ratio of NADPH/NADP ratio, sustaining retinal oxidative stress [19,21,22,23]. Furthermore, increases in oxidative stress dissociate TXNIP from thioredoxin, allowing it to bind other adaptor proteins to activate multiple signaling pathways (reviewed in [14]). For instance, TXNIP has been shown to play a critical role in stress-induced neuronal apoptosis, as it binds reduced Trx and inhibits its activity, releasing free apoptosis signal-regulating kinase 1 (ASK-1) and activating the p38 MAPK and JNK pathway [22,23,24]. In addition, earlier studies showed a close link between IR injury and retinal cell death in general, and specifically neurodegeneration via activation of ER stress markers [25,26,27,28,29]. While we have a good understanding of the events that govern the initial and acute phase of neurodegeneration, the subtle secondary damage of the retina remains not fully understood. In order to devise effective therapeutics, there is a need to better decipher the complex events that contribute to the secondary damage.

In response to virtually all retinal insults, Müller glial cells are activated, evident by enhanced expression of intermediate filaments including GFAP (reviewed in [30]). Our results showed strong gliosis evident by strong radial GFAP staining after 3 days that was sustained for 14 days when compared to shams. Our results lend further support to prior studies showing that IR injury caused Müller cell activation after 1 day [31], 3 days [32], 5 days [33] and persisted for 14 days post-injury [34]. IR-mediated gliosis was paralleled by significant and persistent increases in TXNIP expression. Interestingly, the increase was drastic (7-fold) after 1 day, then (2–3 fold) after 3 days and 14 days. This observation provoked the hypothesis that IR-mediated upregulation of TXNIP might involve different cellular mechanisms. While IR-associated Ca^2+^ influx has been shown to immediately trigger TXNIP expression [7,8], other post-transcriptional mechanisms are likely involved to sustain its expression. We and others have shown that TXNIP can be regulated by miR-17-5p, a small non-coding RNA that inhibits transcription of TXNIP in pancreatic beta-cells [9] and Müller cells [10]. Since colocalization studies revealed that after 3 days, TXNIP was localized within Müller cells, we elected them to investigate TXNIP expression. Several studies have shown that the ER stress and UPR are activated in retinal Müller cells under stress conditions, including oxidative stress, low glucose and hypoxia [35], as well as hyperglycemia [36,37]. Among UPR pathways is IRE1α, an ER bifunctional kinase/RNase that can destabilize miR-17-5p. We found that exposure of Müller cells to hypoxia, followed by reoxygenation to mimic IR injury, was associated with the destabilization of miR-17-5p and significant increases in IRE-1α and TXNIP expression. These findings lend further support to prior findings in pancreatic beta cells [9] and Müller cells [10] in response to metabolic insults. Furthermore, a recent study showed that inhibition of IRE-1α decreased TXNIP expression via preserving miR-17-5p levels using a neonatal hypoxic-ischemic brain injury in rats [38]. Interestingly, deletion of TXNIP normalized the level of ER stress markers and preserved miR-17-5p levels, suggesting a feedback loop of oxidative stress and ER stress in response to IR injury. This effect could be attributed, at least in part, to TXNIP-mediated S-glutathionylation, which has been shown to modulate cell signal [19] and can regulate ER stress and UPR [39]. S-glutathionylation, a posttranslational protein modification, can protect proteins against irreversible oxidative damage, and subsequent deglutathionylation can restore the protein to its native state [40]. Thus, it is conceivable to propose that deletion of TXNIP mitigates elevated S-glutathionylation of protein folding and inhibits expression of ER stress mediators. Future studies are warranted to better understand the molecular events by which TXNIP regulates ER stress.

TXNIP can contribute to retinal inflammation via multiple pathways, including activation of NF-κB and transcription of inflammatory mediators [41], and activation of the inflammasome assembly [36]. Inflammasome is a multiprotein complex of the NLRP that activates an inflammatory cascade by binding procaspase-1 via the caspase recruitment domain of the adaptor protein ASC, wherein then activated caspase-1 can execute proteolytic cleavage of IL-1β [14]. Here, we demonstrated that IR injury-induced significant expression of NLRP3 activated caspase-1 and IL-1β in WT when compared to TKO. To examine the specific role of TXNIP in glial inflammation, we isolated primary Müller cells from both WT and TKO. The results showed that exposure of WT Müller cells, but not TKO to hypoxia/reperfusion, resulted in a significant release of IL-1β into the cell culture medium. These results lend further support to previous reports, showing that TXNIP is integral to inflammasome assembly and release of IL-1β [10,36,42].

Previous literature demonstrated mutual crosstalk between TNF-α and IL-1β, activating the release of each other and exacerbating the UPR response and ER stress [43]. In support of this, our results showed a parallel increase in TNF-α and IL-1β expression in response to IR injury on one hand. On the other hand, knocking down TXNIP expression suppressed NLRP-3 expression and activation, evident by the significant reduction in IL-1β that coincided with decreases in TNF-α expression. Prior work showed a feed-forward loop between ER stress and the pro-inflammatory cytokines, such as IL-1β, IL-6 and TNF-α [44]. The critical role of TXNIP-NLRP3 activation is supported by recent findings showing activation of TXNIP-NLRP3 inflammasome in other models of neurotoxicity [23], critical limb ischemia [16] and stroke and brain injury [38,45].

Here, we observed the activation of Müller glial cells evident by increased GFAP expression that was associated with increases in TNF-α and IL-1β in WT but not in TKO mice. These results are consistent with previous findings, showing that TXNIP plays a pivotal role in glial activation and release of proinflammatory cytokines [10,36,42]. We next examined the effect of the TXNIP-mediated release of inflammatory mediators on secondary damage, including gliosis, neuro and vascular degeneration. As shown in (Figure 5 and Figure 6), retinas from WT, but not from TKO mice, demonstrated significant gliosis and TUNEL-positive cells 3 days post-IR, and increases in vascular cell death (2-fold) 14 days post-IR. In an effort to examine the translational impact of our work, we were able to obtain specific antisense oligomers (ASO) against TXNIP. Intravitreal injection of ASO 48 h post-ischemic injury resulted in a significant reduction in TXNIP mRNA expression at 14 days post-IR. Of note, treatment of the sham tended to increase TXNIP expression when compared to the scrambled-treated sham. However, it did not reach statistical significance. Intervention with TXNIP-ASO after IR injury resulted in significant decreases in the ER stress marker (CHOP) and apoptosis JNK activation when compared to scrambled-treated IR after 14 days (Figure 7). These results not only confirmed the protective effects of genetic deletion of TXNIP, but also suggest that post-injury intervention of TXNIP can modulate biochemical markers of secondary damage, such as in ER stress and apoptotic markers.

Next, we assessed the impact of TXNIP-ASO on the secondary damage at the function level, assessed by post-injury capillary dropout. Having healthy retinal vasculature is critical to maintaining neuronal function. Intervention treatment with ASO resulted in a significant reduction in post-injury capillary dropout, compared with vehicle-treated mice. In support, prior work showed adverse effects of IR injury on retina vasculature via activation of ER stress mediators, including CHOP and IRE1α [46]. Finally, functional consequences of these changes were tested by visual cue behavior studies. Our findings show for the first time that IR injury significantly impaired visual function after 14 days compared with sham controls (Figure 8). Our results show that intervention with ASO exerted functional protective effects in response to IR injury by preventing visual impairment in WT mice. These findings are highly significant and clinically relevant, and could be translated to patients with ocular ischemia. Together, the findings from the current study support the notion that TXNIP plays a critical role in secondary damage post IR-injury. In contrast to the initial phase of neurodegeneration, the subtle secondary damage of the retina involves multiple retina cell types and complex interplay of oxidative stress and ER stress. Here, we demonstrated glial TXNIP expression as a molecular link between oxidative stress and ER stress, as well as a central player in IR secondary inflammation. Further, post-IR intervention with ASO-TXNIP can provide a practical therapeutic window to save visual sight in retinal ischemia.

## 4. Materials and Methods

### 4.1. Animals

All animal experiments were conducted in agreement with the Association for Research in Vision and Ophthalmology statement for use of animals in ophthalmic and vision research, and Charlie Norwood VA Medical Center Animal Care and Use Committee (ACORP#15–04–080). TKO mice were kindly gifted by J. Lusis; they were bred within the facility and crossed with C57BL6-J mice (Jackson Laboratories). These mice were crossed and back-crossed to establish a colony of homozygous TXNIP-/- and WT breeders that produced the mice used in the current study.

### 4.2. Retinal Ischemia-Reperfusion

For surgeries, mice were anesthetized with intraperitoneal ketamine (80 mg/kg; Hospira, Inc., Lake Forest, IL, USA) and xylazine (20 mg/kg; Akorn, Decatur, IL, USA). Retinal ischemia-reperfusion was performed as described previously [18]. Briefly, pupils were dilated with 1% atropine sulfate (Akorn, Inc., Lake Forest, IL, USA). The anterior chamber was cannulated with a 32-gauge needle attached to a line from a saline reservoir at a height calibrated to yield 120 mmHg. The intraocular pressure (IOP) was elevated to 120 mm Hg for 45–60 min; I/R injury and choroidal non-perfusion was evident by the whitening of the anterior segment of the globe and blanching of the episcleral veins [47]. During infusion, topical anesthesia (0.5% tetracaine HCl) was applied to the cornea. After ischemia, the needle was immediately withdrawn, allowing for rapid reperfusion; IOP was normalized, and reflow of the retinal vasculature was confirmed by observation of the episcleral veins. Topical antibiotic was applied to the cornea to minimize infection. IR injury was performed in one eye, with the other undergoing sham surgery, in which the needle was inserted into the anterior chamber without elevating the IOP. Mice were killed 1, 3 or 14 days post-IR and eyes were processed.

### 4.3. Intravitreal Injection of TXNIP Antisense Oligomers (ASO)

Mice were anesthetized by an intraperitoneal injection of ketamine (80 mg/kg)–xylazine (20 mg/kg) mixture and complete anesthesia was confirmed by loss of reflex to sharp paw pinch. WT mice received either TXNIP antisense oligomer or scrambled oligomer (100 µg/2 µL in PBS, Isis Pharmaceuticals). Oligomers were delivered via intravitreal injection 48 h after IR injury using a Hamilton syringe with a 33-gauge glass capillary. Mice were killed after 14 days post-IR and eyes were processed.

### 4.4. Real-Time Quantitative PCR and MicroRNA Detection

Retina samples were processed using a Mirvana PARIS kit and RNA was purified and quantified as described by the manufacturer’s instructions. A one-step quantitative RT-PCR kit (Invitrogen) was used to amplify 10 ng of retinal mRNA as described previously [10]. PCR primers listed in Table 1 were obtained from Integrated DNA Technologies (Coralville, IA, USA). Quantitative PCR was conducted using the StepOnePlus qPCR system (Applied BioSystems, Life Technologies, ). The percent expression of various genes was normalized to 18S and expressed relative to WT sham controls. For micro-RNA detection, a Mirvana PARIS kit (Cat# AM1556, Invitrogen) was used according to the manufacturer’s protocols. Reverse transcriptase reactions, including samples and no-template controls, were run using a TaqMan MicroRNA Reverse Transcription kit (Cat.# 4366596, Applied Biosystems), as described previously [10]. PCR amplification was performed according to the manufacturer’s protocol using TaqMan Universal PCR Master Mix (Cat# 4324018, Applied Biosystems). The percent expression of miR-17-5p was normalized to U6.

### 4.5. Western Blotting

Frozen retinas were placed into protein lysis buffer (Millipore) and briefly homogenized. Retinal lysates were centrifuged and 35 μg were resolved on an SDS-PAGE gel (4–20% gradient Tris glycine precast gel, Bio-Rad) and electro-blotted to nitrocellulose membranes (Bio-Rad). Membranes were blocked with 5% milk or BSA in PBS-Tween, and incubated overnight in 4 °C with one of the primary antibodies listed in Table 2. Membranes were then re-probed with a housekeeping gene, anti-GAPDH or anti-tubulin to confirm equal loading. The primary antibody was detected using a horseradish peroxidase (HRP)-conjugated secondary goat anti-mouse or goat anti-rabbit antibodies and enhanced chemiluminescence. The films were scanned and the band intensity was quantified using densitometry software version 6.0.0 software from alphaEaseFC (Santa Clara, CA, USA) and expressed as relative optical density (OD).

### 4.6. Isolation of Retinal Vasculature and Determination of Occluded (Acellular) Capillaries

The retinal vasculature was isolated as described previously [48]. Freshly enucleated eyes were fixed with 2% paraformaldehyde overnight. Retinal cups were dissected, washed in phosphate-buffered saline, then incubated with 3% Difco-trypsin 250 (BD Biosciences, San Jose, CA, USA) in 25 mmoL/L Tris buffer, pH 8, at 37 °C for 2 h. Vitreous and nonvascular cells were gently removed from the vasculature, which was soaked in several washes of 0.5% Triton X-100 to remove the neuronal retina. Trypsin-digested retinas were stained with periodic acid–Schiff and hematoxylin. Numbers of acellular capillaries were quantified in six different areas of the mid-retina under the microscope (×20) in a masked manner by two different researchers. Acellular capillaries were identified as capillary-sized blood vessel tubes with no nuclei along their length.

### 4.7. Immunostaining of Glial Activation Using GFAP and Colocalization Studies

The distribution of GFAP and colocalization of TXNIP within glial cells in frozen eye sections were analyzed using immuno-histochemistry techniques, as described previously [48]. Retinal sections were fixed with 4% paraformaldehyde, then blocked in goat serum and incubated with primary antibodies listed in Table 3. After removal of primary antibodies, retinal sections were incubated with an appropriate secondary antibody (Table 3). All antibodies were purchased from Invitrogen-Thermo-Fischer Scientific, (Waltham, MA, USA). Specimens were covered with Vectashield mounting medium (Vector Laboratories, Burlingame, CA, USA). Micrographs were taken using a fluorescence microscope (Axiovert-200; Carl Zeiss, Thornwood, NY, USA) at ×20 magnification.

### 4.8. Müller Cell Culture

The rat retinal Müller cell line (rMC-1), a kind gift from Dr. Vijay Sarthy, was utilized to establish the hypoxia reoxygenation model. Primary mouse Müller cells were isolated and used for experiments, as described previously [23]. Briefly, Müller cells were isolated from 6 to 7 day-old mice, and were grown to confluence in complete medium. For the experiments, Müller cells were switched to serum-free media and were exposed to serum starvation for 2 h, followed by 1 h hypoxia (<1% O_2_, 5% CO_2_, 95% N_2_), then 24 h re-oxygenation. Condition medium was collected for enzyme linked immunosorbent assay (ELISA), and cells were harvested for western blot or mRNA quantification.

### 4.9. Slot Blot Analysis

Retinal homogenates (20 µg) were immobilized on a nitrocellulose membrane, as described previously [49]. After blocking, membranes were reacted with antibody against 4-hydroxynonenal (4-HNE) from Calbiochem (San Diego, CA, USA), followed by HRP-conjugated sheep anti-rabbit antibody and enhanced chemi-luminescence (GE Healthcare). The optical density of various samples was quantified using densitometry software version 6.0.0 software from alphaEaseFC (Santa Clara, CA, USA) and expressed as optical density (OD).

### 4.10. ELISA

Levels of IL-1β in cell-conditioned media were detected by IL-1β ELISA sensitive kits (R&D Systems, Minneapolis, MN, USA). Equal volumes of conditioned media for each treatment group were concentrated using Amicon 10K concentration columns (Millipore, Temecula, CA, USA) and then the ELISA was performed by following the manufacturer’s protocol. The levels of IL-1β were expressed as pg·mL−1 of cell-conditioned media.

### 4.11. Visual Assessment

Visual function, including visual acuity and pattern discrimination tasks, were assessed behaviorally by training and testing mice on the ‘cue’ version of the Morris water maze task, as described previously [17]. Animals were placed for 10 s on a platform in a tank of opaque water, 22–25 °C, which was elevated above the water surface (2 cm) and clearly visible from any location in the tank. Subsequently, there were four trials per day for 3 days or until a stable performance plateau was reached. On each trial, animals started from different locations at the periphery of the tank and were allowed to swim to the escape platform. If they did not reach the platform in 60 s, they were gently guided to it by the investigator. They remained on the platform for 10 s. Visual impairment was diagnosed if the experimental group had a higher escape time compared to either its own pre-surgery level, or compared to the post-surgery control group escape times.

### 4.12. Terminal dUTP Nick End-Labeling (TUNEL) Analysis

Eyes were mounted in OCT and sections (10 μm) were collected and stored at −80 °C. TUNEL was performed in these frozen sections using the Apop-TAG in situ cell death detection kit (TUNEL-FITC; Chemicon International) as described previously [23]. Retina sections were cover-slipped with Vectashield containing DAPI (Vector Laboratories, Burlingame, CA, USA). Micrographs were captured at 20× by fluorescent microscope (AxioObserver.Z1; Zeiss, Jena, Germany).

### 4.13. Quantification of Total Neuronal Cells in Ganglion Cell Layer (GCL)

Retinal sections containing the optic nerve were fixed using 2% paraformaldehyde and cover-slipped with Vectashield (Vector Laboratories, Burlingame, CA, USA). Images were captured by fluorescent microscope (AxioObserver.Z1; Zeiss, Jena, Germany). The number of cells in the ganglion cell layer (GCL) was counted, except for blood vessels in two areas around the optic nerve and in two areas in periphery of one ora serrata to the other, as described previously [50]. Cells were counted in a masked manner by two independent investigators.

### 4.14. Statistical Analysis

All the data are expressed as mean ± SEM. Differences between two groups were detected using an un-paired Student’s *t*-test. One-way ANOVA was used to assess the significant differences between three groups. Two-way ANOVA was used to assess the interaction between two variables, two genes (WT vs. TKO) and (IR vs. sham). Tukey–Kramer post-multiple comparisons was used for significant interactions among various groups. Significance for all tests was determined at α = 0.05, Graph-pad Prism, Ver.6.

## 5. Conclusions

Together, the findings from the current study support the notion that TXNIP plays a critical role in secondary damage post IR-injury. In contrast to the initial of phase of neurodegeneration, the subtle secondary damage of the retina involves multiple retina cell types and a complex interplay of oxidative stress and ER stress. Here, we demonstrated glial TXNIP expression as a molecular link between oxidative stress and ER stress, as well as a central player in IR secondary inflammation. Furthermore, post-IR intervention with ASO-TXNIP can provide a practical therapeutic window to save visual sight in retinal ischemia.

## Figures and Tables

**Figure 1 ijms-20-03969-f001:**
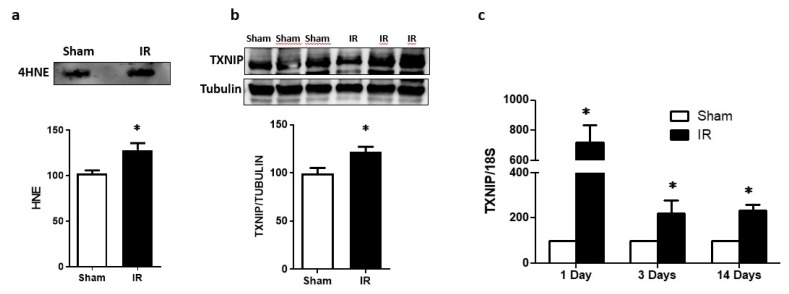
(**a**) Ischemia/reperfusion (IR) induced oxidative stress assessed by slot-blot analysis of 4-HNE level compared to sham-control after 1-day (*n* = 3). (**b**) Representative western blot and statistical analysis showed significant increase in TXNIP expression 1day post-IR (*n* = 3). (**c**) Statistical analysis of mRNA level assessed by realtime-PCR showed that IR induced the expression of TXNIP in WT mice after 1-day that was sustained for 3 and 14 days post-IR (*n* = 4). * *p* < 0.05 vs. sham.

**Figure 2 ijms-20-03969-f002:**
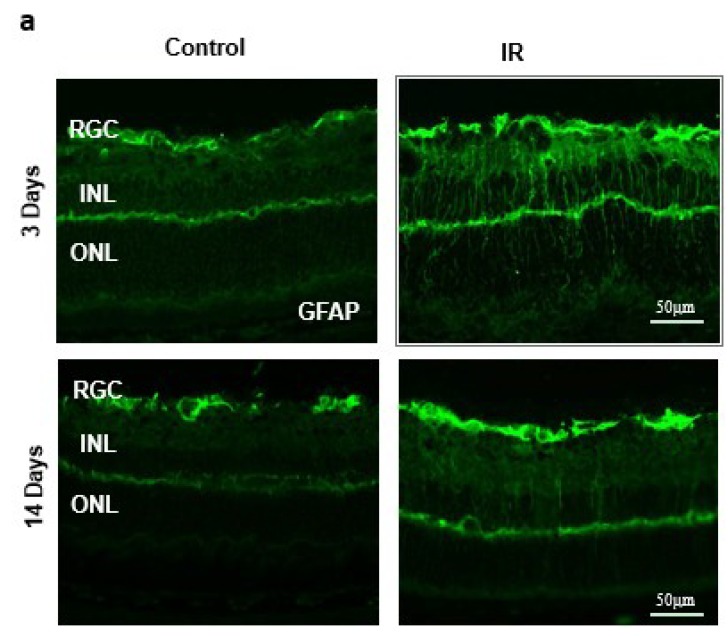
(**a**) Ischemia/Reperfusion (IR) induced strong activation of Muller cells, the main glia in the retina, assessed by GFAP radial staining after 3-days and persisted for 14-days after ischemia. (**b**) Immunostaining studies using anti-TXNIP (green), anti-glutamine synthetase (GS, red) showed prominent colocalization (yellow) of TXNIP within Muller cells in response to IR when compared to shams.

**Figure 3 ijms-20-03969-f003:**
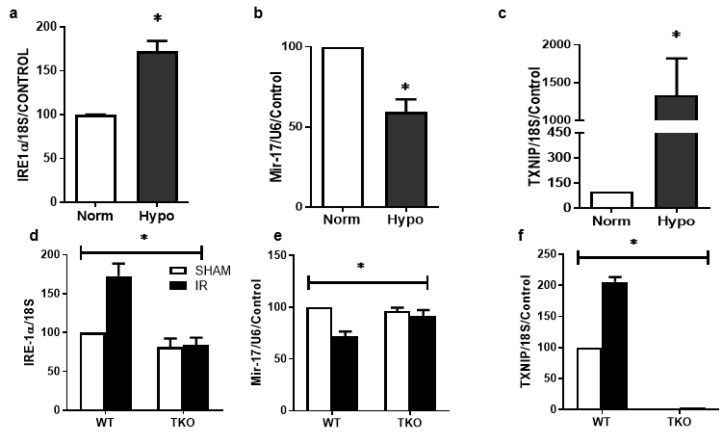
Rat Muller cells (rMC-1) were exposed to hypoxia for 1hour followed by reperfusion in normoxia for 24-hours. Statistical analysis of mRNA level assessed by realtime-PCR showed a significant increase in ER-stress marker IRE-1α (**a**), decrease in miR-17-5p (**b**), Marked increase (20-fold) in TXNIP mRNA (**c**) compared to cells grown in normoxia. (*p* < 0.05 vs. normoxia, *N* = 4–6). Deletion of TXNIP blunted the increase in IRE-1α (**d**), restored miR-17-5p and increase in TXNIP (f) in response to IR when compared to WT (*N* = 4–6, * *p* < 0.05, two-way ANOVA two-way ANOVA showed significant effect of IR and gene deletion).

**Figure 4 ijms-20-03969-f004:**
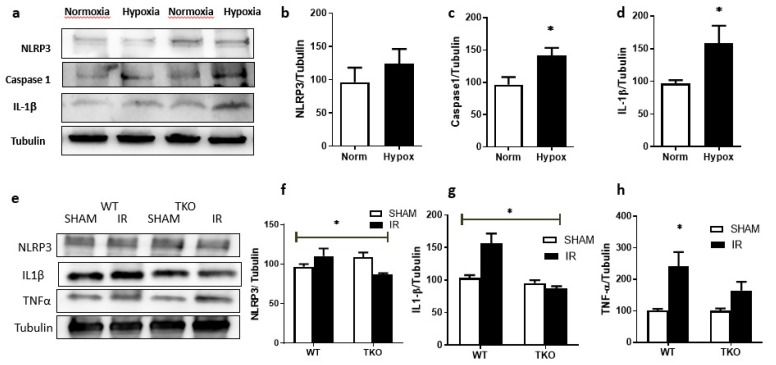
Primary Muller cells isolated from wildtype mice were exposed to hypoxia for 1hour followed by reperfusion in normoxia for 24-hours. (**a**) Representative of Western blot analysis and Statistical analysis showed no significant difference in NLRP3 expression (**b**), significant increase in Caspase-1 (**c**), and IL-1β (d) (* *p* < 0.05 vs. normoxia, *N* = 4–6). (**d**) Representative of Western Blot analysis and statistical analysis showed that deletion of TXNIP blunted the increase in NLRP3 (**e**), IL-1β (**f**) and increase in TNF-α (**g**) in response to IR when compared to WT (*N* = 4,* *p* < 0.05, two-way ANOVA showed significant effect of IR and gene deletion).

**Figure 5 ijms-20-03969-f005:**
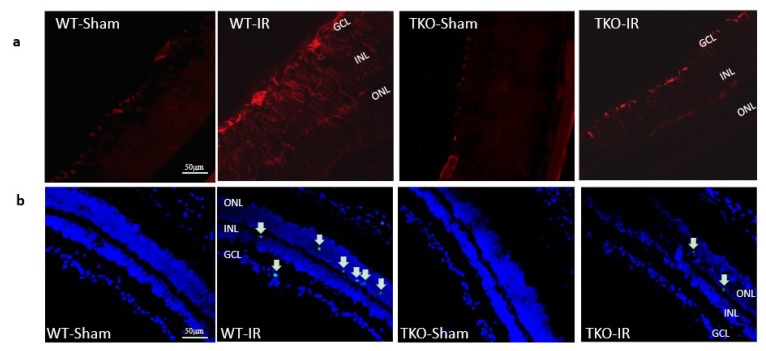
(**a**) Ischemia/Reperfusion (IR) induced strong activation of Muller cells, the main glia in the retina, assessed by GFAP radial staining after 3-days in WT-IR but not in TKO-IR when compared to shams. (**b**) Ischemia/Reperfusion (IR) sustained retinal cell death indicated by TUNEL-positive cells (arrows) after 3-days in WT-IR, but not in TKO-IR when compared to shams.

**Figure 6 ijms-20-03969-f006:**
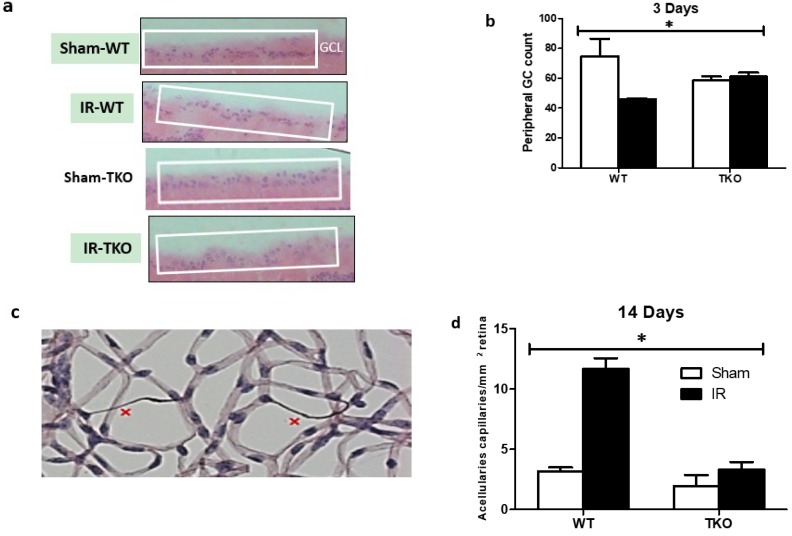
(**a**,**b**) Representative images and statistical analysis showing IR caused significant reduction in total number of cells in ganglion cell layer (GCL) in the peripheral retinas of the WT, but not of TKO mice when compared to shams after 3-days. (**c**,**d**) Representative image and statistical analysis showing IR significantly increased development of occluded (acellular) capillaries in WT, but not TKO mice when compared to shams after 14-days. (*n* = 8–10, * *p* < two way ANOVA showed disease and gene interaction, * *p* < 0.05).

**Figure 7 ijms-20-03969-f007:**
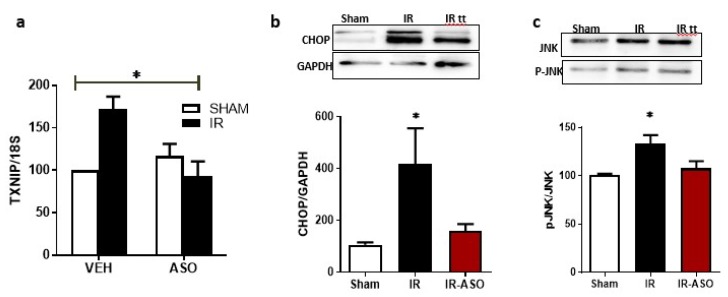
(**a**) Statistical analysis of mRNA level assessed by realtime-PCR showed that that interventional treatment with antisense oligomer (ASO); a novel specific TXNIP inhibitor; blunted the increase of TXNIP mRNA 14-days post IR-injury. (* *p* < 0.05 two-way ANOVA showed significant interaction of IR and treatment, *N* = 3–4) (**b**,**c**) Western blot analysis showed that interventional treatment with TXNIP-ASO blunted the increase of the ER-stress marker, CHOP and the apoptotic marker Phosphorylated-JNK expression after retinal IR. (* *p* < 0.05 vs. sham, *n* = 3–4).

**Figure 8 ijms-20-03969-f008:**
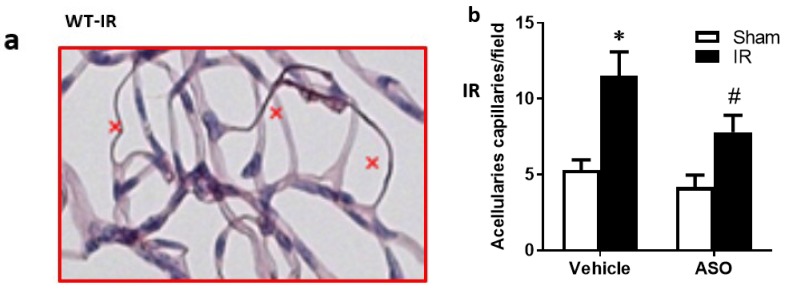
(**a**,**b**) Representative image and statistical analysis showing that transient ischemia increased number of acellular capillaries (marked with x), which was reduced by inhibiting TXNIP expression 2 days after IR. (**c**,**d**) I/R led to an increase in escape time, inhibiting TXNIP reduced the time to reach the platform 6 and 9 days after injury. (*n* = 8–10, two-way ANOVA showed significant disease and treatment interaction, * *p* < 0.05 ASO vs. IR vehicle).

**Table 1 ijms-20-03969-t001:** The sequence of the PCR primers used in the mRNA quantification experiments.

Gene	Forward	Reverse
**18 S**	CGCGGTTCTATTTTGTTGGT	AGTCGGCATCGTTTATGGTC
**IRE1** **α**	GGGTTGCTGTCGTGCCTCGAG	TGGGGGCCTTCCAGCAAAGGA
**TXNIP**	AAGCTGTCCTCAGTCAGAGGCAAT	ATGACTTTCTTGGAGCCAGGGACA

**Table 2 ijms-20-03969-t002:** List of antibodies and sources used to detect protein expression by Western Blot.

Antibody	Source	Catalogue #	Company
**TXNIP** **TXNIP**	Monoclonal Polyclonal	K0205-3 403700	MBL Abacus ALS Australia Invitrogen-Thermo-Fischer Scientific, Waltham, MA
**NLRP-3**	Polyclonal	LS-B4321	LifeSpan Biosciences, Inc, Seatle, WA
**IL1β**	Polyclonal	ab9722	Abcam, Cambridge, MA
**TNF-a**	Polyclonal	ab9635	Abcam, Cambridge, MA
**Tubulin**	Monoclonal	ab4074	Abcam, Cambridge, MA
**GAPDH**	Polyclonal	5174	Cell Signaling Tech, Danvers, MA
**CHOP**	Polyclonal	3082	Cell Signaling Tech, Danvers, MA

**Table 3 ijms-20-03969-t003:** List of antibodies and sources used to detect protein expression by Immunohistochemistry.

Antibody	Source	Catalogue #	Dilution
**TXNIP**	Polyclonal	403700	1:100
**Glutamine Synthetase (GS)**	Monoclonal	MA5-27749	1:100
**Glial Fibrillary Acidic Protein (GFAP)**	Polyclonal	PA5-16291	1:200
**Oregon green-Conjugated secondary antibody**	Goat anti-rabbit	O-11038	1:500
**Texas red-Conjugated secondary antibody**	Goat anti-mouse	T-862	1:500

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
