# Peer review of "Modulating Expression of Thioredoxin Interacting Protein (TXNIP) Prevents Secondary Damage and Preserves Visual Function in a Mouse Model of Ischemia/Reperfusion"

_ijms, 2019, doi:10.3390/ijms20163969_

Round 1

Reviewer 1 Report

The authors have addressed the reviewers’ comments by re-writing some passages of the manuscript and by editing several figures.

However, there are still several items that need to be corrected.

The methods section is missing a paragraph on how the retinal ganglion cells were counted. There should be a section: “histology”. Also, To clearly differentiate retinal ganglion cells form displaced amacrine cells, it would have been better to use a RGC marker by immunohistochemistry or to define size criteria for counting the cells.

Specific comments:

Introduction, p.2, line 51: “neural stem cell” should be plural (neural stem cells).

Results:

p.3, line 135: “Although TUNEL+ve cells were detected in retinal ganglion cell (RGC) layer, inner nuclear layer (INL), the majority of cells …” – there should be an “and” between “… (R“… GC), and inner nuclear …”

p.4, line 142: Text should refer to Fig. 6a, b

p. 4, line 146: Text should refer to Fig. 6 c, d

p. 4, line 165: Text should refer to Fig. 8a, b

p. 4, line 169: Text should refer to Fig. 8c

p. 4, line 171: Text should refer to Fig. 8d.

Figures: Figures 2, 5, 6, 7 and Supplementary Figures 2, 3 should have calibration bars.

Figure 8a: the red Xs are difficult to see.

The RGC images in Figure 6A and Supplementary Figure 2 are taken from the same sections. However, there is a difference in label. What is labeled IR-WT in Figure 6, looks like sham-TKO in Supplementary Figure 2. What is labeled Sham-TKO in Figure 6, looks like IR-WT in Supplementary Figure 2. What is it? It would have been better to show images with high magnification. The differences in the images are not that convincing.

Author Response

The methods section is missing a paragraph on how the retinal ganglion cells were counted. There should be a section: “histology”. Also, to clearly differentiate retinal ganglion cells form displaced amacrine cells, it would have been better to use a RGC marker or to define size criteria for counting the cells.

 We do apologize for this oversight and the method section is now updated with the criteria for counting the cells as suggested. We are familiar with markers that differentiate RGC from displaced Amacrine cells (Mohaed et al 2018, PMID: 29253516), however, the plan was to identify total loss by counting the number of neuronal cells in the ganglion cell layer post-IR injury.

Specific comments:

Introduction:

p.2, line 51: “neural stem cell” should be plural (neural stem cells). Corrected as suggested.

Results:

p.3, line 135: “Although TUNEL+ve cells were detected in retinal ganglion cell (RGC) layer, inner nuclear layer (INL), the majority of cells …” – there should be an “and” between “… (R“… GC), and inner nuclear …” Corrected as suggested.

p.4, line 142: Text should refer to Fig. 6a, b Corrected as suggested

p. 4, line 146: Text should refer to Fig. 6 c, d Corrected as suggested

p. 4, line 165: Text should refer to Fig. 8a, b Corrected as suggested

p. 4, line 169: Text should refer to Fig. 8c Corrected as suggested

p. 4, line 171: Text should refer to Fig. 8d. Corrected as suggested

Figures:

Figures 2, 5, 6, 7 and Supplementary Figures 2, 3 should have calibration bars. 

Inserted as suggested

Figure 8a: the red Xs are difficult to see. 

An enlarged window of the original image is now shown. The exact window is identified in the original figure in supplementary figure-3.

The RGC images in Figure 6A and Supplementary Figure 2 are taken from the same sections. However, there is a difference in label. What is labeled IR-WT in Figure 6, looks like sham-TKO in Supplementary Figure 2. What is labeled Sham-TKO in Figure 6, looks like IR-WT in Supplementary Figure 2. What is it? It would have been better to show images with high magnification. The differences in the images are not that convincing.

The GCL images in Fig.6A  are taken from the full images shown in Supplementary Fig.2. and we have verified that each representative in Fig.6A came from the corresponding group shown in supp Fig.2. The other point is that the data was generated by counting multiple retina sections from multiple animals and the differences are subtle.  

To make it easier on the reader, In the revised figure, a white box that identified which area is taken from the full image  is now shown (supl Fig.2). 

Reviewer 2 Report

Authors have addressed all my concern. One humble suggestion, authors should safely  secure all the samples before acceptance of manuscript. So, whenever reviewer have some concerns  they can be addressed accordingly. 

Author Response

On behalf of the authors, we would like to thank the reviewer for taking time and reviewing our manuscript. We appreciate the critiques and suggestions.

Round 2

Reviewer 1 Report

The authors have improved the figures and the text according to the reviewers’ suggestions.

The have clarified how they counted the cells in the ganglion cell layer. Since they counted all cells (including displaced amacrine cells), p. 3, line 141 should not state “ganglion cell count” but rather “count of cells in the ganglion cell layer”.

Author Response

p. 3, line 141 should not state “ganglion cell count” but rather “count of cells in the ganglion cell layer”.

Corrected as suggested and your thoroughness is greatly appreciated.

This manuscript is a resubmission of an earlier submission. The following is a list of the peer review reports and author responses from that submission.

Round 1

Reviewer 1 Report

The purpose of this study was to investigate the role of Thioredoxin interacting protein (TXNIP) in retinal ischemia/reperfusion, and in a tissue culture model (Müller cells) of hypoxia. Ischemia/reperfusion and hypoxia was associated with upregulation TXNIP and stress-related proteins, increase of the inflammasome, activation of retinal Müller cells, and downregulation of miR-17-5. All these effects could be eliminated by deletion of TXNIP expression (either TXNIP-KO mice (TKO) or treatment with antisense oligonucleotides targeting TXNIP (ASO). The effect of ASO treatment was also shown in a behavioral test (watermaze).

In general, the manuscript is well written with good experimental design. However, there are some issues that need to be addressed.

Specific comments:

Results:

p. 3, line 118: there is a reference to Fig. 5A about the upregulation of NLRP3 expression in retinas from wildtype but not in TKO-mice. This should be a reference to Fig. 4A.

p. 3, line 140: The text should refer to Fig. 6A,C.

p. 4, line 142-143: The text should refer to Fig. 6B, D.

p. 4, line 150: There should also be a description of the results in Fig. 7 B,C.

Methods:

p. 7, line 300-301: The ketamine dose is referred to as 237.7 g/mol, 100mg/ml, similar for the Xylazine dose (220.3 g/mol, 100 mg/ml). This is the concentration of ketamine and Xylazine in the vial. The authors should just list the dosage in mg/kg.

p. 8, line 343: The authors should give the concentrations used for anti GFAP, and for the secondary antibodies.

Figure 5: The TUNEL-positive cells are very difficult to see because of the strong DAPI. It would be better to also show the single green channel (without blue).

Figure 6A): The images appear a bit fuzzy (perhaps due to the pdf), and have too low magnification. It would be helpful to also show higher magnification of the ganglion cell layer (perhaps as inserts).

Figure 8A): too low magnification to see anything. Proposal: Enlarge this panel to the height of the current figure, and place the diagrams on the right side.

Reviewer 2 Report

In this manuscript, the authors have examined the Modulating Expression of Thioredoxin Interacting Protein(TXNIP) Prevents Secondary Damage and Preserves Visual Function in A Mouse Model of Ischemia/ Reperfusion. Manuscript is very well written and results are convincing too. Authors have shown the importance TXNIP protein IR injury and its possible role in clinical trials. Experimental design is very good. Material methods are well explained. I have some concerns. After addressing these changes manuscript can be accepted.

Concerns:

1.   In material methods section 4.4. real time PCR-Are authors sure they used 10 ng RNA.

2.   It will be nice if authors can show DCF or DHE staining along with 4-HNE that will justify the role of accentuated ROS in IR model.

3.   Authors should do experiments related to NADPH/NADP ratio either in cells or in mice. This will provide better understanding of role of TXNIP as NADPH is the major cofactor used by thioredoxins.

4.   In Figure 1 A label the slot blot.

5.   It is little surprising how come the expression of IRE1alpha in TKO sham is same as control sham group and then after IR it does not increases, which it should not be. As far my understanding goes to begin in TKO expression of IRE-1a should be less so that further increase is prohibited under IR conditions. Please provide better graph or repeat the experiment.

6.   In figure 4a please provide better blot for caspase-1 blot. And 4 b IL1B seems to be more in TKO sham then IR, according to study plan it should be other way around.

7.   In figure 7 a with ASO TXNIP should be less in sham group then only further down regulation is possible in IR group. Explain this or repeat the experiments.